

# Parked and Operating Loads Analysis in the Aerodynamic Design of Multi-megawatt-scale Floating Vertical Axis Wind Turbines

Mohammad Sadman Sakib[1] and D. Todd Griffith[1]

[1]The University of Texas at Dallas, Richardson, TX, 75080, USA

**Correspondence:** D. Todd Griffith (tgriffith@utdallas.edu)

**Abstract.** A good understanding of aerodynamic loading is essential in the design of vertical axis wind turbines (VAWTs) to properly capture design loads and to estimate the power production. This paper presents a comprehensive aerodynamic design study for a 5 MW Darrieus offshore VAWT in the context of multi-megawatt floating VAWTs. This study systematically analyzes the effect of different, important design variables including the number of blades ($N$), aspect ratio ($AR$) and blade tapering in a comprehensive loads analysis of both the parked and operating aerodynamic loads including turbine power performance analysis. Number of blades ($N$) is studied for 2- and 3-bladed turbines, aspect ratio is defined as ratio of rotor height (H) and rotor diameter (D) and studied for values from 0.5 to 1.5, and blade tapering is applied by means of adding solidity to the blades towards blade root ends, which affects aerodynamic and structural performance. Analyses were carried out using a three-dimensional vortex model named CACTUS (Code for Axial and Crossflow TUrbine Simulation) to evaluate both instantaneous azimuthal parameters as well as integral parameters, such as loads (thrust force, lateral force, and torque loading) and power. Parked loading is a major concern for VAWTs, thus this work presents a broad evaluation of parked loads for the design variables noted above. This study also illustrates that during the operation of a turbine, lateral loads are on par with thrust loads, which will significantly affect the structural sizing of rotor and platform & mooring components.

## 1 Introduction

In order to improve the current situation of global climate change there is a push for more environmentally sustainable sources of energy in the power grid. As wind energy costs have come down, the opportunity to fill this need by advancing the potential of wind turbine technology is at an all-time high as a part of the initiative to reduce greenhouse gas emissions and provide greater generation by renewable sources. The wind energy community has been mostly focused on Horizontal Axis Wind Turbines (HAWTs), but recently another type of turbine named Vertical Axis Wind Turbines (VAWTs) (Bhutta et al. (2012); Sutherland et al. (2012)) have garnered interest especially in offshore locations (Griffith et al. (2018); Möllerström et al. (2016)). VAWTs have some distinct advantages over HAWTs because of their low center of gravity (c.g), absence of pitch and yaw mechanism, directional independence to wind and low sound emission (due to low operational TSR) (Griffith et al. (2018); Möllerström et al. (2016)).

VAWTs received significant amount of attention during the 1970s and 1980s in the USA and Canada (Paraschivoiu (2002)). Some commercial VAWTs were developed, but they suffered from several problems (namely, fatigue and low efficiency). In



the meantime HAWT research gained momentum and it became the industry standard for multi-megawatt scale wind energy conversion systems. Despite these issues, interest in VAWTs have resurfaced especially in offshore deep-water conditions and it has been discovered VAWTs might challenge HAWTs in terms of efficiency and cost-effectiveness (Simão Ferreira (2009); Griffith et al. (2018); Ennis and Griffith (2018)).

As wind energy is moving to deployments in deeper waters offshore, maintenance and installation procedures as well as cost trends are changing. It has also been shown that floating VAWTs have the potential to achieve a significant reduction in the cost of energy (COE) compared to floating HAWTs (Shelley et al. (2018); Griffith et al. (2016)). In addition, floating VAWTs are much more suitable in wind farm conditions due to quicker dissipation of wakes when placed in counter rotating pairs (Kinzel et al. (2012)). Comparative studies of HAWTs and VAWTs have been performed in detail to identify the advantages

and disadvantages of each design concept in offshore and onshore conditions (Islam et al. (2013); Henderson and Patel (2003)). As a result, interest in floating VAWTs is re-surging, and various floating VAWT concepts have been proposed, including the DeepWind concept (Paulsen et al. (2012)), VertiWind concept (Tjiu et al. (2015)), etc.

  In order to make VAWTs a viable candidate as a replacement for HAWTs, one must have reliable aerodynamic models, and reliable predictions of aerodynamic design loads and power production. Some well-established aerodynamic models include

the momentum models (Single Stream tube (SST) model, Multiple Stream tube (MST) model, Double Multiple Stream tube (DMST) model) and Vortex Models (Islam et al. (2008)). In the momentum models Bernoulli's equation is used for each stream tube to calculate flow velocity and forces. The vortex models are essentially potential flow models with free vortex line elements. Rotor blades are modelled using the lifting line approximation, with each blade discretized into a number of blade elements containing a bound vortex and whose strength are obtained utilizing airfoil coefficients (i.e.; airfoil polars). More

details on aerodynamic modelling and their fidelity can be found in (Ferreira et al. (2014); Cheng et al. (2016); De Tavernier et al. (2020)).

  A major concern for VAWTs is cyclic aerodynamic loading throughout one revolution, which gives rise to serious issues like high root blade bending moments and fatigue (Galinos et al. (2016)). One of the major reasons justifying the lack of commercial VAWTs is that the fatigue loading in VAWTs has been a challenge for component reliability under these cyclic

loads; however, it has been discovered that those issues were exacerbated primarily due to early VAWTs using aluminum blade materials (Möllerström et al. (2019)) rather than high fatigue-resistant composites. Another major concern for VAWTs are the loading on blades during standstill (parked) conditions. As VAWTs are omni-directional and typically do not have a pitching mechanism they are subjected to high loads for a long period of time. Parked load studies for VAWTs have been reported to a very limited extent, especially for Darrieus-type VAWTs. For this reason we place in this work a particular emphasis on parked

loads calculations.

  The aim of this work is to establish and better understand the aerodynamic design process of a floating 5MW Darrieus VAWT and how the selection of various design variables affect this design process without necessarily going into the detailed explanation of how these design variables affect the flow physics. This study includes blade and tower design loads, rotor power performance, and the loads that are imparted to the floating platform and mooring system. In order to find an efficient design

we perform a comprehensive aerodynamic performance and loads analysis by studying trade-off between different, important





design variables. These variables include the number of rotor blades ($B$), blade chord tapering ($\beta$) and the aspect ratio ($AR$). Attention has been brought to the parked loads and lateral loads issue which are typically not studied during the design process.

Simulations are carried out using a 3D vortex-based code named CACTUS (Murray and Barone (2011)) to check both instantaneous azimuthal parameters, as well as integral parameters, such as thrust coefficient and power coefficient. This mid-fidelity tool provides a good balance between accuracy and computational cost, which is of great importance to efficiently sweep through large design spaces. The results of this tool will provide a clear picture of the design space and the importance of making appropriate choices for designing a realistic VAWT.

In summary, the research objective is twofold:

1. Studying the impact of important design variables like the number of rotor blades ($B$), blade tapering ($\beta$) and the aspect ratio ($AR$).

2. Providing an understanding of the VAWT's cyclic loads as well as the parked loads, which are typically not investigated.

## 2 Numerical Model and Method for VAWT Aerodynamics

CACTUS is a 3D aerodynamic design code capable of performing an analysis of arbitrary turbine configurations (Murray and Barone (2011)). The code was based on VDART3, a free vortex wake simulation of the Darrieus wind turbine, developed by Strickland et al. (1979). The rotor blades are represented by the lifting line approximation. The blades are discretized into several elements and and to each element, two-dimensional lift and drag data for any desired airfoil is assigned. At each point in time, the vortex line structure is composed of trailing and spanwise wake vorticity, and a bound vortex system attached to the blade elements. The bound vorticity on each element is related to the element lift coefficient through the Kutta-Joukowski theorem (Anderson Jr (2010)), and the trailing and spanwise wake vorticity are recovered through the application of the Helmholz theorem (Katz and Plotkin (2001)). The wake convection velocity is either calculated based on the induced velocity at every time step (free-wake) or kept constant in time (fixed-wake). A vortex core model is included to avoid instabilities near the vortices. As the aerodynamics of cross-flow turbines is inherently unsteady, the blades are subjected to dynamic loading and often operate at angles of attack beyond their steady-state stall limits for long period of time. This transient behavior is called dynamic stall effects and is incorporated in CACTUS by two dynamic stall models. The modified Boeing-Vertol method by Gormont (1973) and the Leishman-Beddoes model by Leishman and Beddoes (1989). The coordinate frame used in this study is presented in Fig. 1. More details on CACTUS and validation against experimental turbines can be found in Murray and Barone (2011); Michelen et al. (2014); Wosnik et al. (2016); Lu (2020).

Following the results of convergence studies, each blade in the model for a 5MW Darrieus VAWT is represented by 10 elements and 30 time steps are used per rotor revolution. Calculations are run for 10 rotor revolutions with free-wake and with no dynamic stall model, which leads to a maximum of 0.75 % difference between the last two revolution averaged power coefficient at high tip speed ratios (TSR) and a maximum of 0.005% difference at low tip speed ratios (TSR).



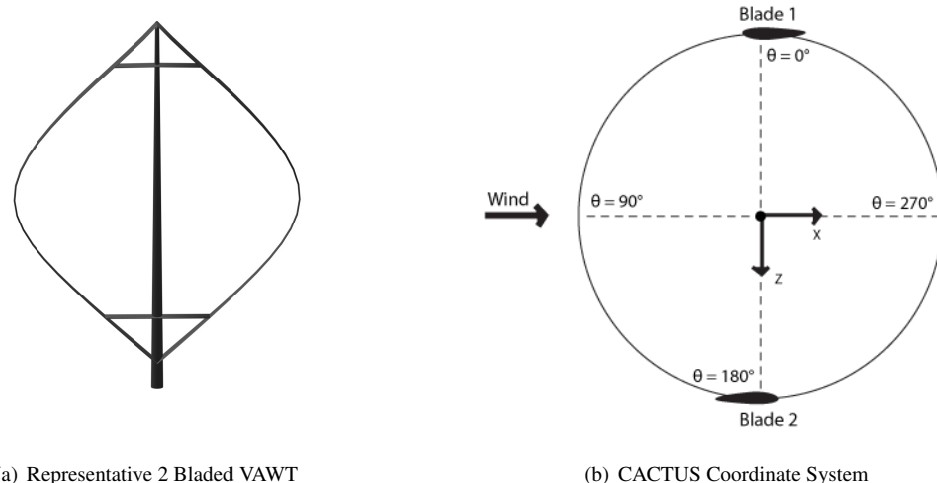

(a) Representative 2 Bladed VAWT      (b) CACTUS Coordinate System

**Figure 1.** Schematic view of a 2 bladed VAWT and CACTUS coordinate system.

## 2.1 Test Cases

The present work presents a design study aiming to reveal the impact of different important design variables on the aerodynamic performance of VAWTs and subsequently how it affects the design of other major components of the turbine or floating system.

Therefore the parameters are systematically studied which will provide a comprehensive picture of the influence of design parameters under various operational conditions.

In order to evaluate the relative cost of candidate rotor designs, annual energy production must be estimated. Further, aerodynamic loads are required for iterative design of the turbine blades and platform design. As mentioned previously, CACTUS is used to perform all the necessary analysis.

For the purpose of this study, the VAWT machines are assumed to be stall-regulated, with no active power or loads control. They are also assumed to operate with a simple variable-speed controller to optimize energy capture in below-rated (Region 2) of the power curve. A cut-in, cut-out and rated wind speed of 5 m/s, 25 m/s and 15 m/s respectively, are imposed. The performance data presented here are generated assuming zero wind shear; only small quantitative differences are expected when shear is introduced, and these are not expected to impact the resulting trends in performance with design changes. For

aerodynamic modeling purposes, all rotor designs employ NACA-0021 airfoils at all blade sections and will be simulated for the following operational conditions (Density ($\rho$)= 1.225, Viscosity ($\mu$) = $1.790243.10^5$, Temperature (T) = 293K). Finally the revolution averaged performance data for power coefficient ($C_P$) and torque coefficient ($C_Q$) at both low and high TSR are shown in Fig. 2 to show required convergence of results.

Six Darrieus rotor designs were analyzed for the blade tapering and number of blade study, each design incorporating a

unique combination number of blades (2 and 3), and choice of blade tapering ($\beta$ = 0, 0.5 and 1). These six rotor designs have a fixed blade radius of 54m, a rotor frontal area of $9513.15m^2$ and a fixed aspect ratio (AR) of 1.22. The degree of





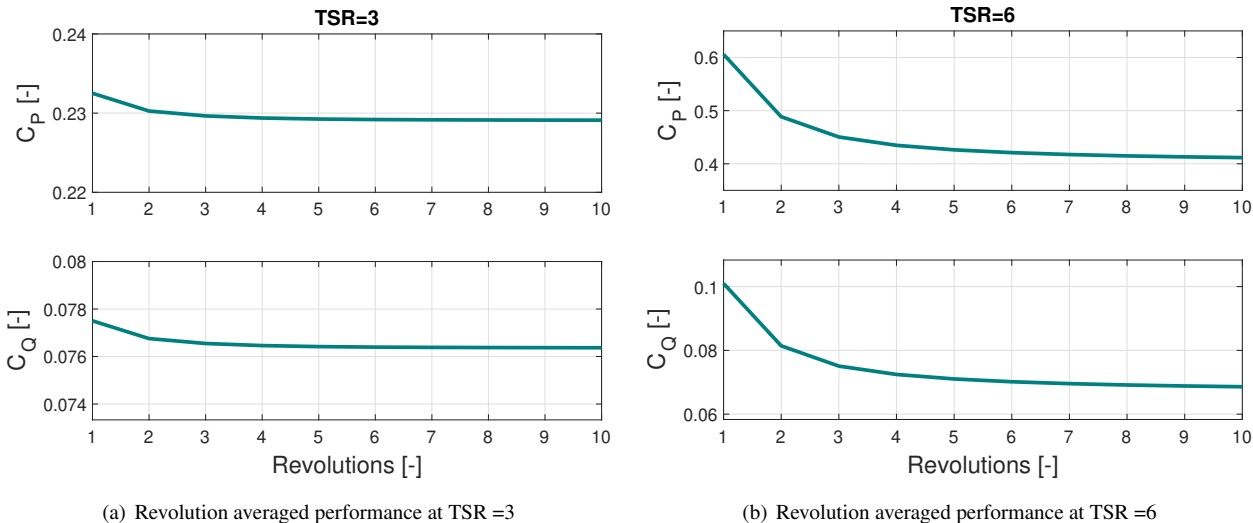

(a) Revolution averaged performance at TSR =3      (b) Revolution averaged performance at TSR =6

**Figure 2.** Revolution averaged performance for 10 revolutions at (a) low TSR and (b) high TSR.

tapering is denoted by a non-dimensional parameter called $\beta$ ranging from 0-1. $\beta$ value of 0 means no tapering (i.e.; constant blade chord) and a value of 1 indicates the highest degree of tapering has been applied to the blades with increasing chord toward the blade root connection to the tower. Tapering is applied by adding solidity to the blades which means that the design

configurations with lower $\beta$ values have a lower overall solidity. A visual representation of the tapering scheme is shown in Fig. 3. The geometry has been generated in such a way that the solidity remains the same for the 2-bladed (2B) and 3-bladed (3B) configuration with the same $\beta$ value. In this study the aspect ratio is defined as the ratio between rotor height (H) and diameter (D). For the aspect ratio study four different values have been selected ranging from (0.5-1.5) for a particular blade number and blade tapering scheme while keeping the rotor radius same. The ratio between rotor height (H) and diameter (D) is being

changed in such a manner that the rotor diameter remains constant. A summary of the rotor design candidates is presented in Table 1.

**Table 1.** Rotor design candidate summary.

| Number of Blades ($B$) | Blade Chord Tapering ($\beta$) | Maximum Chord ($c$) [$m$] | Aspect Ratio ($H/D$) [$-$] | Radius [$m$] |
|---|---|---|---|---|
| 2 | 0, 0.5, 1 | 3, 4.5, 6 | 0.5,0.75,1,1.5 | 54 |
| 3 | 0, 0.5, 1 | 2, 3, 4 | 0.5,0.75,1,1.5 | 54 |



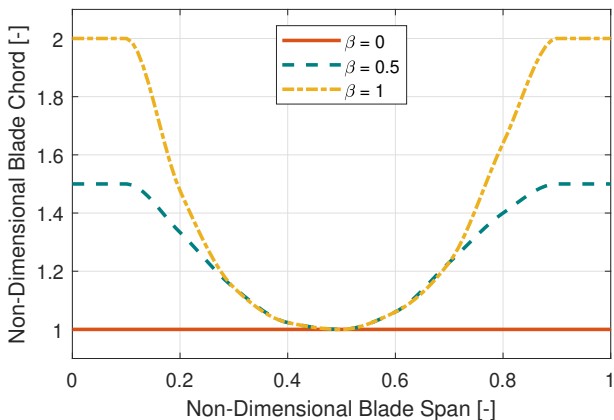

**Figure 3.** A visual representation of the tapering scheme.

### 2.1.1 Operational Strategy Development

The first step of the aerodynamic analysis is the development of the operational strategy as a function of wind speed. To generate the power curve, operating thrust and lateral loads of each design case, a (Wind Speed[WS]-RPM) schedule must be

defined that is assumed to be the operational condition of the turbine. As a first step, a power coefficient $C_P$- TSR curve is generated as shown in Fig. 4(a). During this step a reasonable estimation of the rated RPM value is made. From this $C_P$-TSR curve, the maximum $C_P$ and the TSR at which the turbine stalls is extracted as shown in Fig. 4(b). These two values are used in the second stage of the operational strategy development.

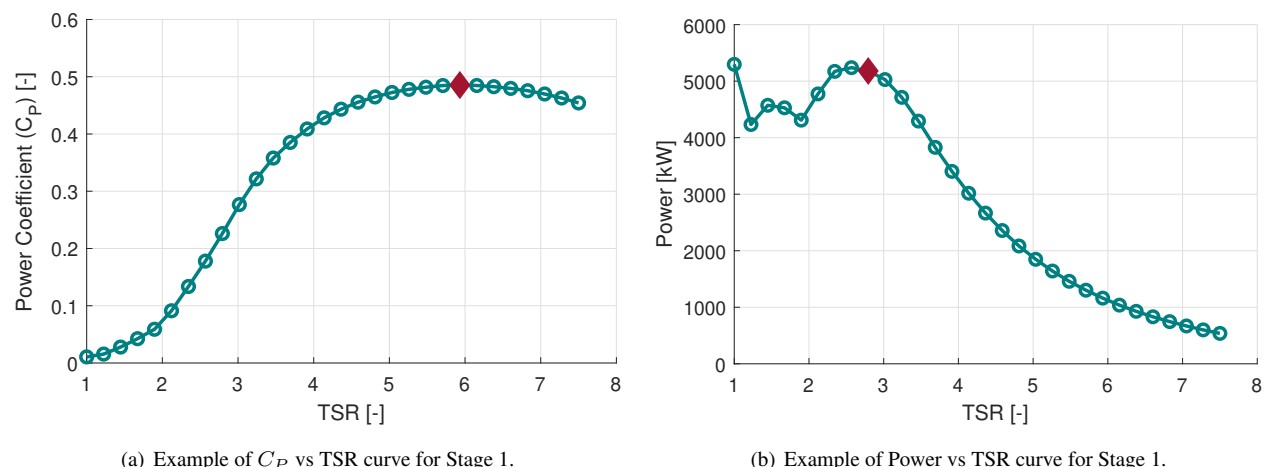

(a) Example of $C_P$ vs TSR curve for Stage 1.

(b) Example of Power vs TSR curve for Stage 1.

**Figure 4.** Examples of a $C_P$ and Power vs. Tip Speed Ratio curve used for stage 1 of the analysis. The maximum $C_P$ and the stall point are indicated by red markers.



In the second stage, a RPM sweep is performed at the stall TSR to find the required RPM to produce the desired rated
aerodynamic power of 5 MW at the selected rated wind speed of 15 m/s. An example curve is shown in Fig. 5(a) where the
rated RPM is 7.8.

In the third stage, the rotor RPM schedule is completely defined as a function of wind speed. The RPM is set to the minimum
of the RPM giving the optimal tip speed ratio (TSR) for a given wind speed, and the RPM for rated power at stall. This defines
a simple variable speed control schedule that provides for optimal energy capture under the constraint that the required RPM
to maintain rated power is never exceeded. An example of an RPM schedule and resulting operating TSR range is shown in
Fig. 5(b).

This operational strategy is used to generate all the operational loads and power information including power curves; and the
cyclic thrust, lateral, and torque loads vs. wind speed for all the rotor design candidates. As a result the AEP can be calculated
from the power curve whereas the thrust, lateral and torque loads provide the maximum design loads and the cyclic fatigue
loads used to design the turbine and the platform and mooring system.

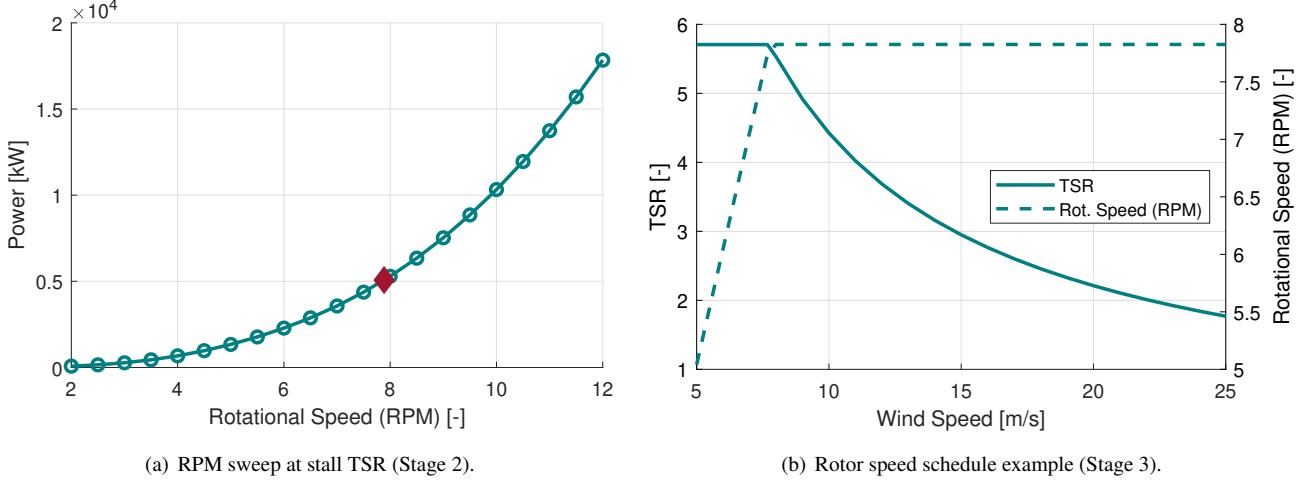

(a) RPM sweep at stall TSR (Stage 2).  (b) Rotor speed schedule example (Stage 3).

**Figure 5.** Stage 2 and 3 of the operational strategy development.

## 3 Results for the Aerodynamic Design Study

The impact of blade tapering, number of blades and the aspect ratio on the performance of a turbine and their impact on the
design process is discussed in detail in this section.





## 3.1 Case 1: Impact of Blade Chord Tapering

In this study tapering is applied to blades by adding solidity, as shown in Fig. 3. Solidity is an important parameter in the design of a VAWT as it has a massive impact on the aerodynamic performance. Solidity is defined as

$$\sigma = Bc/R \qquad (1)$$

where $B$ is the number of blades, $c$ is the blade chord and $R$ is rotor radius. The effect of solidity on turbine performance has been studied extensively in both numerical and experimental studies. Rezaeiha et al. (2018) studied the effect of solidity on

turbine performance ( power Coefficient ($C_P$) and thrust Coefficient ($C_T$) ) and optimum TSR using 2D CFD simulations. This study showed that an increase in solidity caused an increase in the power coefficient ( due to an increase in Reynolds Number), and the optimum TSR value decreases while the thrust coefficient ($C_T$) grows asymptotically. Miller et al. (2018) studied the effect of solidity on VAWT performance at high Reynolds number experimentally . Their study concluded that with an increase in Reynolds number the power coefficient increases and shows asymptotic behavior above 1.5 million Reynolds number. More

work on the effect of solidity both experimentally and numerically can be found in (Maeda et al. (2016, 2017); Delafin et al. (2016); Jafari et al. (2018)).

Tapering provided various advantages and disadvantages over conventional non-tapered blades. As tapering adds solidity to the blades it will have a higher performance coefficient ($C_P$) compared to non-tapered blades while keeping the blade mass increase under acceptable levels. In addition to that tapered blades with variable chord along the span should be more

structurally sound than non-tapered blades due to a larger chord towards the blade roots. VAWTs experience very high blade root moments (Galinos et al. (2016)); therefore, placement of more stiffness towards the blade root ends via larger root chords will result in better strength and fatigue life while having acceptable blade mass increase. But adding tapering also leads to higher parked loads; a major concern in VAWTs and this is studied further in this work. Also increasing solidity or adding tapering will also reduce maximum RPM as increase in solidity also increases the aerodynamic torque, which increases the

cost of drivetrain.

### 3.1.1 Turbine Performance

Fig. 6 shows the power coefficient versus tip speed ratio for 2-bladed (2B) and 3-bladed (3B) VAWTs with three different tapering schemes ($\beta$ = 0, 0.5 & 1). Each of these rotors has a radius of 54m and an aspect ratio (AR) of 1.22. It can be seen that the power coefficient increases with $\beta$ value due to increase in solidity. The trend of optimum TSR value can also be

seen, where for a higher degree of tapering the optimum TSR decreases and vice-versa. It is to be noted that 2 and 3-bladed configurations with the same $\beta$ value have similar solidity for the whole blade span. A rotor can be considered well-designed if the power coefficient is relatively high for a longer range of TSR rather than having a very high value for only a fraction of the same range of TSR. As VAWTs in this study and in general are stall controlled, during majority of its operation it will be operating at sub-optimal TSR values so it's important to have a good power coefficient ($C_P$) value for a larger range of TSR.

Fig. 7 shows the power curves for the 6 design candidates and the maximum $C_P$ and maximum design RPM for each candidate is provided in Table 2. Increasing the tapering, comes at the price of lowering the maximum RPM, which results in

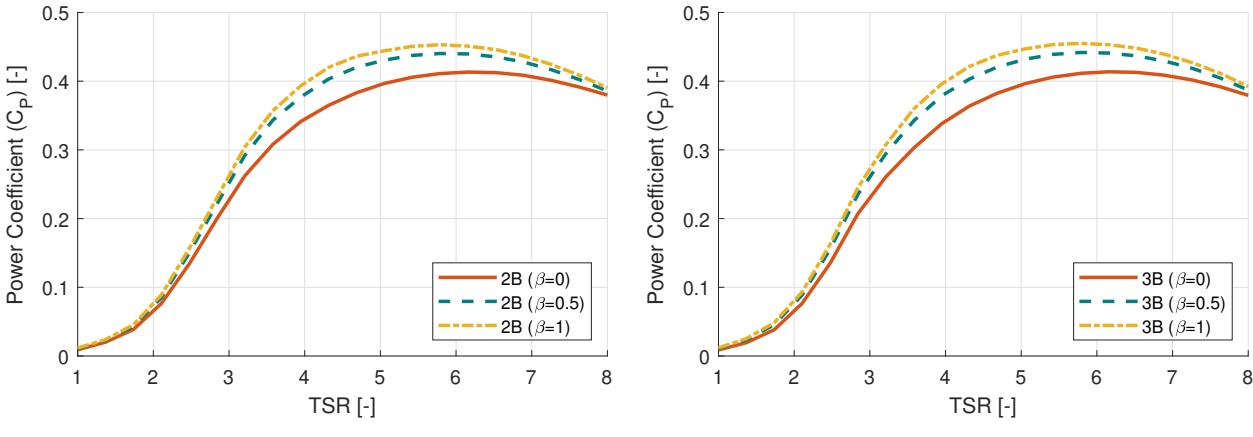

(a) Power coefficient ($C_P$) vs TSR curve for 2-bladed (2B) rotors.   (b) Power coefficient ($C_P$) vs TSR curve for 3-bladed (3B) rotors.

**Figure 6.** Rotor power coefficient ($C_P$) analysis.

higher torque that increases the cost of the drivetrain. The power curves were generated according to the operational strategy development as discussed earlier. As the 2- and 3-bladed configurations with the same $\beta$ value have similar solidity, then as a result the power curves for those corresponding designs are almost similar prior to stall but they differ drastically post stall.

That is because rotors with higher beta value (larger tapering) will have higher tip losses post stall and that results in lower power production. As a result low $\beta$ rotors have higher power production compared to high $\beta$ rotors in the post-stall region.

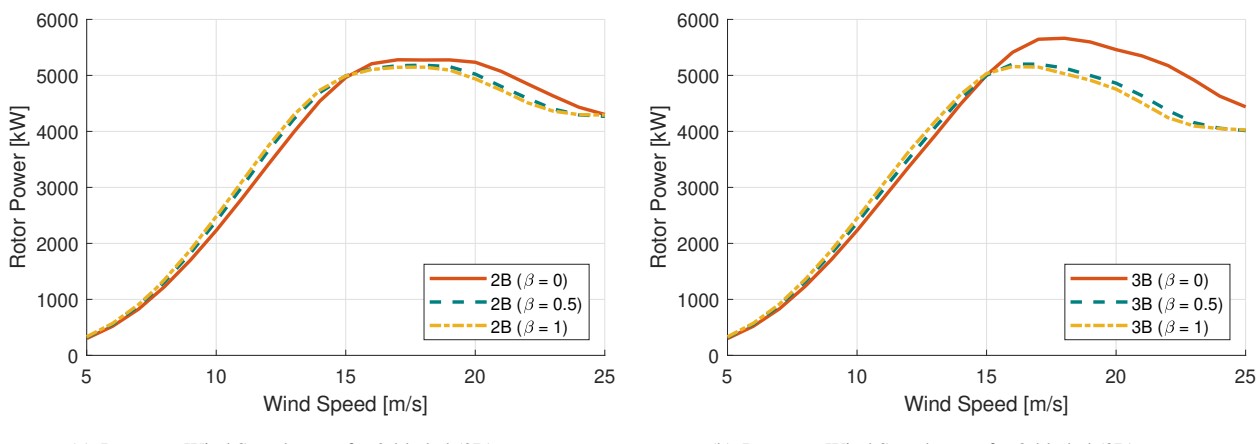

(a) Power vs Wind Speed curve for 2-bladed (2B) rotors.   (b) Power vs Wind Speed curve for 3-bladed (3B) rotors.

**Figure 7.** Rotor power performance analysis.





**Table 2.** Rotor Design Candidate Summary

| $\beta$ [−] | Max Rotor $C_P$ 2B | Max Rotor $C_P$ 3B | Max Rotor RPM 2B | Max Rotor RPM 3B |
|---|---|---|---|---|
| 0 | 0.4133 | 0.4137 | 8.33 | 8.38 |
| 0.5 | 0.4403 | 0.4418 | 7.95 | 7.82 |
| 1 | 0.4531 | 0.4551 | 7.82 | 7.69 |

### 3.1.2 Turbine Loads

The thrust ($F_X$) and lateral force ($F_Z$) are important design considerations for the structural design of the platform, blades and tower of a floating wind turbine. Fluctuations of these forces generate fatigue loads and may therefore increase the capital cost. This section presents the thrust and lateral force coefficients trends for the design candidates. In the existing literature there is a lack of investigation of the lateral loads and how they behave with changing TSR. The thrust ($F_X$) or lateral force ($F_Z$) is dimensionalized as

$$F_{X/Z} = \frac{1}{2}\rho U_\infty{}^2 A_T C_{x/z} \tag{2}$$

Where $\rho$ is the air density, $U_\infty$ is the freestream velocity, $A_T$ is the frontal area and $C_x$ is the required thrust load coefficient and $C_z$ is the lateral load coefficient. With an increase in blade tapering and TSR, the value of the mean thrust coefficient ($C_x$) over a single revolution increases as shown in Fig. 8. But turbines with a similar solidity ( for example 2B ($\beta = 0$), 3B ($\beta = 0$)) show similar profiles for thrust coefficient ($C_x$). With an increase in blade tapering the chord value of the blades increases which creates larger loads on the turbine. A similar study has been performed for the lateral load coefficient ($C_z$). The mean, maximum and minimum of the lateral load coefficient ($C_z$) at each TSR have been plotted in terms of bar plots for each design configuration in Fig. 9. It can be seen that the mean lateral loads coefficient is independent of blade tapering or solidity and values are close to zero. However, the lateral loads are almost symmetric in nature and have very high amplitudes ranging from almost equal negative values to positive values. An increase in solidity or blade tapering causes the absolute value of maximum and minimum lateral load coefficients to increase as well. With increasing TSR the loads continue to increase in a uniform manner for the 2-bladed turbine while for the 3-bladed turbine at low TSR we see an increase in amplitude at the beginning, and again starts following similar trend to the 2-bladed turbine when it is not operating at low TSR. It is to be noted that there is also a significant cyclic variation in loading for thrust coefficient ($C_x$) too. But Fig. 8 only reports the mean thrust coefficient ($C_x$) over a single revolution and the bars have been omitted to reduce complexity of the plot. The cyclic variation in both thrust and lateral loading have been discussed in detail in the impact of number of blades section.

### 3.1.3 Parked Loads

The forces on a turbine during parked conditions at extreme wind speeds are a major load case to investigate, to ensure safe operation of the turbine. Parked loads are of increasing concern because VAWT design loads tend to be maximum for the



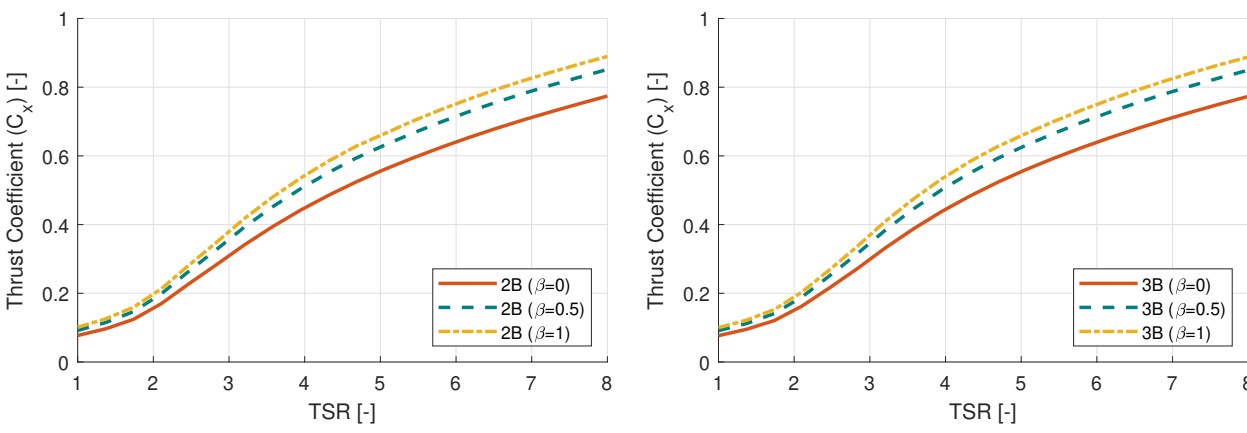

(a) Thrust coefficient ($C_x$) vs TSR curve for 2-bladed (2B) rotors.  (b) Thrust coefficient ($C_x$) vs TSR curve for 3-bladed (3B) rotors.

**Figure 8.** Rotor thrust coefficient ($C_x$) analysis.

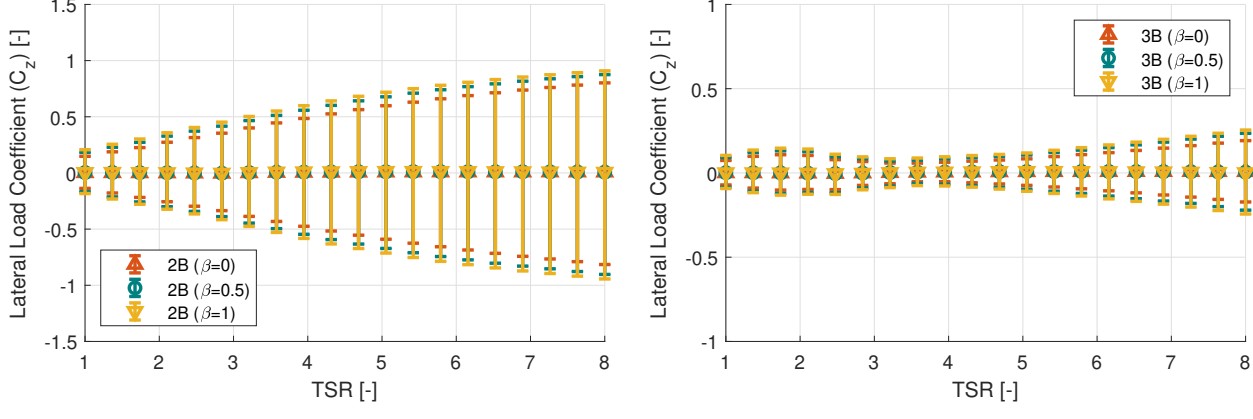

(a) Lateral load coefficient ($C_z$) vs TSR curve for 2-bladed (2B) rotors.  (b) Lateral load coefficient ($C_z$) vs TSR curve for 3-bladed (3B) rotors.

**Figure 9.** Rotor lateral load coefficient ($C_z$) analysis.

cut-out wind speed, and the loads increases with wind speed (wind force relation) due to the absence of a pitching mechanism (Griffith et al. (2018)).

Despite the importance of parked loads in VAWTs, very few studies have been conducted on the parked loads calculations
of VAWTs, thus in this section we present parked loads for various levels of blade chord tapering, and in subsequent sections we examine the impact of other parameters on parked loads. Ottermo et al. (2012) used analytical expressions to estimate storm/extreme loads at parked conditions to find an optimal parking strategy. Experimental measurements of parked loads on blades were conducted on a 12 kW VAWT by Goude et al. (Goude and Rossander (2017)). Finally some computational fluid dynamics (CFD) on operating loads and parked loads were done by Paulsen et al. (2013) for the DeepWind concept.





In this section, parked loads for the 6 design candidates will be studied to understand the effects of design variables starting with blade tapering on the parked loads. As methods like CFD will be very computationally intensive to predict parked loads, a simplified analytical method has been developed. Some of the assumptions of this study are that, blade loads are calculated from measured 180 deg lift and drag data for the NACA-0021 profile provided with CACTUS. As the turbine is parked, static airfoil data has been used. No wake effects have been considered, one blade does not affect another. Also shadow effects (blade,

tower), strut effects, finite aspect ratio corrections have not been considered.

    The numerical model computes the solution for a 3D Darrieus rotor, for a set of specified azimuthal positions, horizontal planes and blade elements similar to CACTUS. As a first step the relative velocity encountered by the blade is determined assuming that the rotational component and the disturbance velocity induced by wake and bound vorticity is zero. Once the relative velocity is found the instantaneous angle of attack ($AOA$) can be determined. It is to be noted that the normal and

tangential vectors attached to each blade element are considered while calculating local blade element angle of attack, and thus is essential to include the blade curvature effects for a 3D Darrieus rotor. Once the angle of attack ($AOA$) at each azimuthal position for each blade element at specific horizontal planes are known one can find the instantaneous lift and drag forces on the blade elements from static airfoil data. These lift and drag forces are resolved into normal ($C_n$) and tangential ($C_t$) components of forces. Once the element normal and element tangential components are known the forces in wind direction (thrust load,

$F_X$) and perpendicular to wind direction (lateral load, $F_Z$) on a turbine scale can be calculated. A flowchart detailing the steps involved in parked loads calculation is shown in Fig. 10.

    A comparison of the numerical simulation has been performed where the results are compared with Ottermo et al. (2012) as shown in Fig. 11. The peak values match very well , but there are some differences in the overall shape of the profiles. This arises due to the difference in overall theory behind calculation of parked loads for both studies.

There are applicable standards in extreme meta-ocean condition like the 50-year and 100-year return periods. As International Electrotechnical Commission (IEC) prefers 50-year return periods for extreme design conditions, in this work the parked loads are calculated using a site specific 50-year return period having and 10 minute average wind speed of 30.94 m/s with a shear exponent of 0.11 (Sirnivas et al. (2014)).

    We now examine the rotor thrust and lateral parked loads for different blade chord tapering with results shown in Fig. 12

for parked thrust-direction loads and in Fig. 13 for parked lateral-direction loads. In both Fig. 12 and Fig. 13 the results for 2-bladed rotors are plotted on the left and compared with results for 3-bladed rotors on the right. As expected with the increase of tapering the maximum value of the thrust loads increases for both 2B and 3B configurations. This trend is also observed for lateral parked loads where the absolute value of the maximum load also increases with tapering.

## 3.2   Case 2: Impact of Number of Blades

The number of blades is an important design consideration, which significantly affects the aerodynamic performance of a VAWT because a change in the number of blades will significantly affect the forcing frequency of the cyclic aerodynamic loads. Further, the number of blades may cause a change in solidity, Reynolds number, blade wake interaction, stall behavior etc. The effect of changing the number of blades on aerodynamic forces generated in VAWTs and the performance coefficients



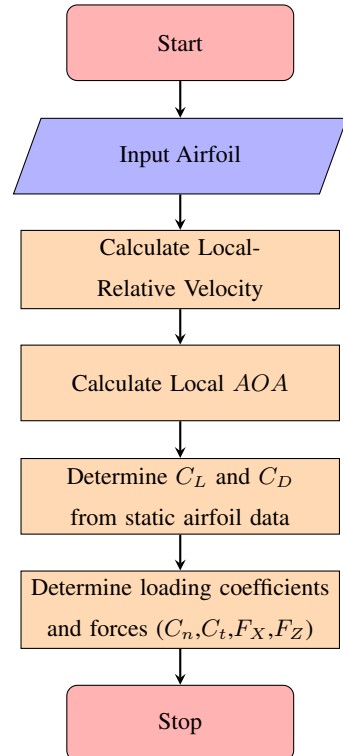

**Figure 10.** Flowchart of parked loads numerical model.

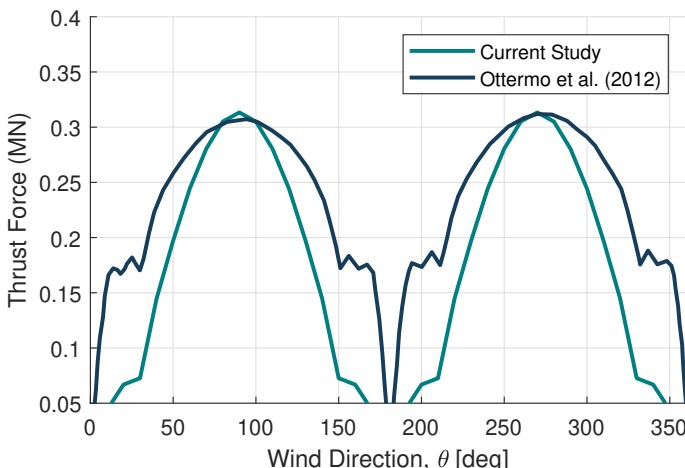

**Figure 11.** Parked loads comparison with existing literature.

will be discussed in this section. This work investigates the impact of number of blades on the steady and cyclic turbine loads

including thrust loads, lateral loads, parked loads and power ripple effects of Darrieus turbines.





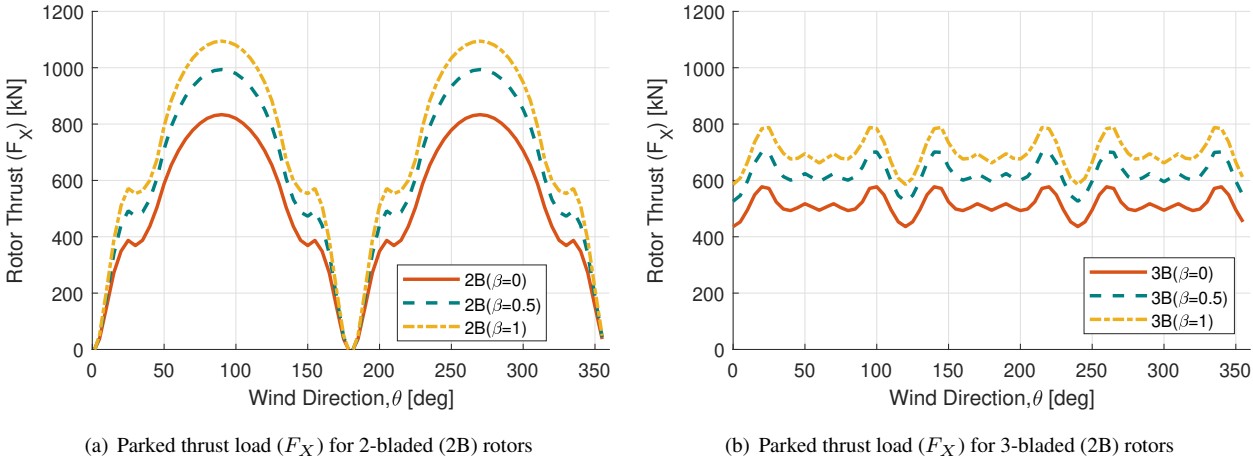

(a) Parked thrust load ($F_X$) for 2-bladed (2B) rotors      (b) Parked thrust load ($F_X$) for 3-bladed (2B) rotors

**Figure 12.** Parked thrust load ($F_X$) analysis for 2-bladed and 3-bladed rotors.

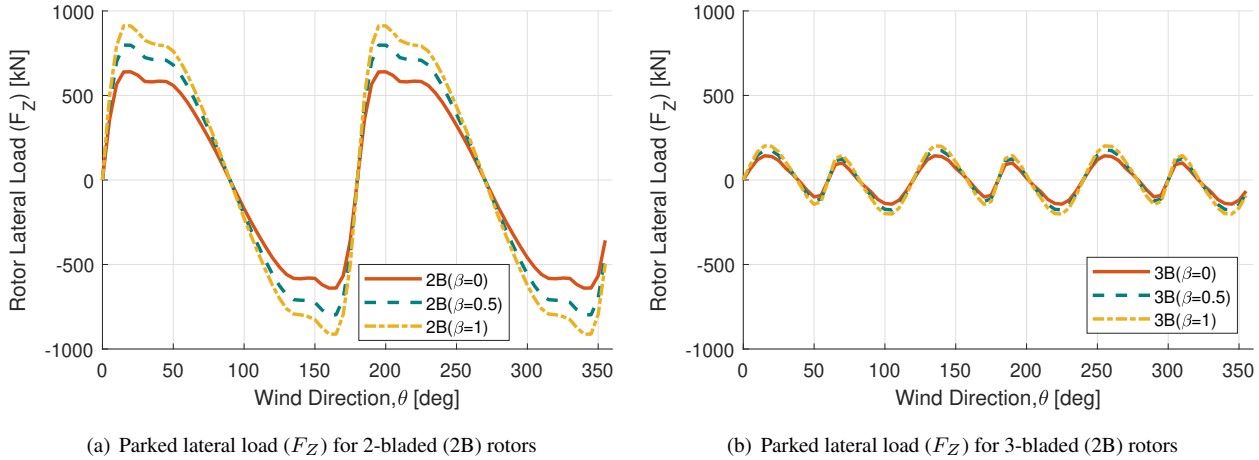

(a) Parked lateral load ($F_Z$) for 2-bladed (2B) rotors      (b) Parked lateral load ($F_Z$) for 3-bladed (2B) rotors

**Figure 13.** Parked lateral load ($F_Z$) analysis for 2-bladed and 3-bladed rotors.

The influence of number of blades on VAWT aerodynamics have been studied previously in detail; however, most studies have focused primarily on VAWTs with straight blades, which are known as H-VAWTs. Rezaeiha et al. (2018) studied the optimal aerodynamic design of H-VAWTs through a systematic study of solidity and number of blades with the help of 2D CFD simulations. The study indicates that increasing the number of blades has no effect on the averaged power coefficient, while it does impact the uniformity of the power output and loading. Castelli et al. (2012) also studied the effect of the number of blades on H-VAWTs using 2D CFD simulations. The study reveals that with increasing number of blades the power coefficient decreases, but the study was set up in such a way that the solidity is not maintained for all design considerations. As solidity has not been kept constant across all designs this might lead to erroneous observation and cannot truly isolate the effect of number of blades on performance. Delafin et al. (2016) studied the effect of number of blades on Darrieus rotors using






a free wake vortex model similar to this study. The study concludes that with an increase in the number of blades the torque, the thrust and the lateral ripple effects are significantly reduced. Some experimental work also has been done regarding blade number effects. Maeda et al. (2015) studied the effect of number of blades (ranging from 2 to 5) of a H-VAWT in wind tunnel. Power coefficient, torque coefficient and thrust coefficient against TSR were compared for the range of blade numbers (2 to 5). The study shows that with an increase in blade number the power coefficient decreases and torque coefficient increases. The measurements also show that as blade number increases the thrust coefficient increases as well, but this is due to the fact the solidity increases with the addition of a blade.

### 3.2.1 Turbine Performance

To demonstrate the impact of the number of blades on turbine performance, the power coefficient ($C_P$) versus TSR and solidity for the six rotor configurations is studied (similar to Fig. 6) for both low wind speed and high wind speed conditions. It is seen from Fig. 14 that for low wind speed (5m/s) the 2-bladed rotors have slightly better aerodynamic performance than 3-bladed rotors. This is primarily due to Reynolds number effect which is dependent on blade chord value and relative blade velocity. As the 2-bladed turbines have higher chord values than 3-bladed turbines they are subjected to higher Reynolds number which leads to slightly better aerodynamic performance. The noticeable difference mainly arise in low TSR region where the blade is operating in dynamic stall conditions and operating Reynolds number is lower compared to high TSR regions as seen from Fig. 14(b) which is a zoomed in plot of Fig. 14(a). And as the TSR increases the difference between 2-bladed and 3-bladed performance becomes almost identical as Reynolds number is sufficiently high for both turbines. Similar study has been performed at high wind speed (14.5m/s) where the Reynolds number is considerably higher for all TSR values and this study is similar to the one done in section 3.1. As for this case, the power coefficient ($C_P$) is mostly invariable to change in number of blades for all TSR values. So it can be considered that the power coefficient ($C_P$) is insensitive to the change in number of blades which confirms that selection of number of blades should be done based on performance of other design variables rather than the $C_P$.

### 3.2.2 Turbine Loads

Fig. 15 shows the trend of the thrust coefficient ($C_x$) for the six rotor configurations in the same figure to observe the impact of number of blades. This study is similar to Fig. 8 of section 3.1.3 where the rotor thrust coefficient ($C_x$) versus TSR were plotted separately for 2-bladed and 3-bladed rotors. There are minor differences in thrust coefficient ($C_x$) at low TSR regions and these differences are even smaller at high TSR where the Reynolds number for both 2-bladed and 3-bladed turbines are very high. A similar conclusion can be reached for rotor thrust coefficient ($C_x$) which is; they are insensitive to number of blades provided that the solidity is kept constant across rotor configurations with different blade number.

We now turn our attention to cyclic aerodynamic load variations where the thrust and lateral load profiles for 2- and 3-bladed turbines are shown in Fig. 16. To quantify the change in cyclic load amplitude range ($F_{X/Z}^{max} - F_{X/Z}^{min}$) of the load profiles are computed similar to other study by Delafin et al. (2016). Adding another blade to a 2-bladed turbine reduces the cyclic load amplitude by 85.2 %. Similarly for the lateral load profiles the cyclic load amplitude decreases by 83.8 %. For similar solidity



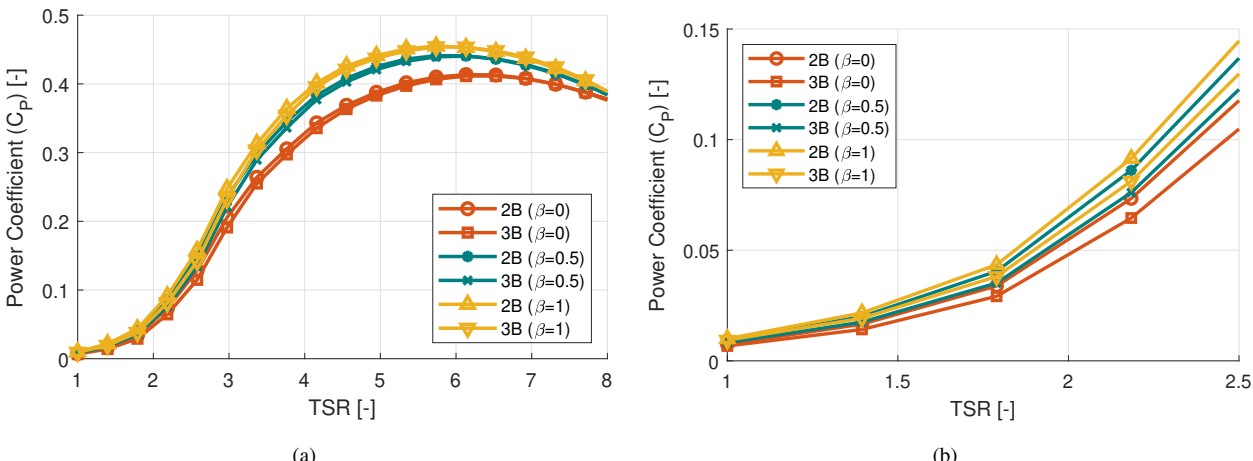

**Figure 14.** Power coefficients ($C_P$) versus TSR for 2-bladed and 3-bladed rotor at low wind speed (5 m/s).

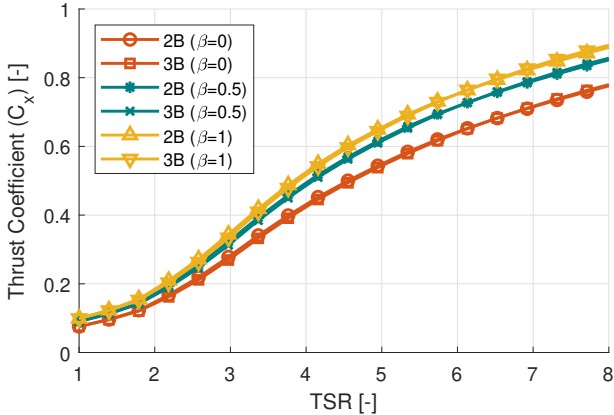

**Figure 15.** Parked loads comparison with existing literature.

turbines the average value of the load profiles are very similar, the 2B ($\beta$=0) and 3B ($\beta = 0$) have a mean thrust value of 449.9 kN and 428.9 kN respectively. For lateral loads the mean values of the 2B($\beta$=0) and 3B($\beta = 0$) rotors are 3.615 kN and 3.572

295  kN. Increasing the blade number from 2B to 3B changes the phase of the load profiles and also adds a 3P component to the load profiles. These changes have significant impact on the selection of other design elements like the turbine generator and tower design, blade mass and fatigue life in the rotor and floating system. Next comes the question of how the mean, maximum and minimum value trend for the full operating range of wind speeds (5-25m/s) look like? The profile of thrust and lateral loads for the full operating range of wind speeds is shown in Fig. 17 where a significant reduction in cyclic load amplitude for

300  the 3-bladed rotor is evident for both thrust and lateral loads across all wind speeds. Further, it can be noted that the maximum (peak) operating loads for the 3-bladed rotor are also significantly lower than for the 2-bladed rotor.





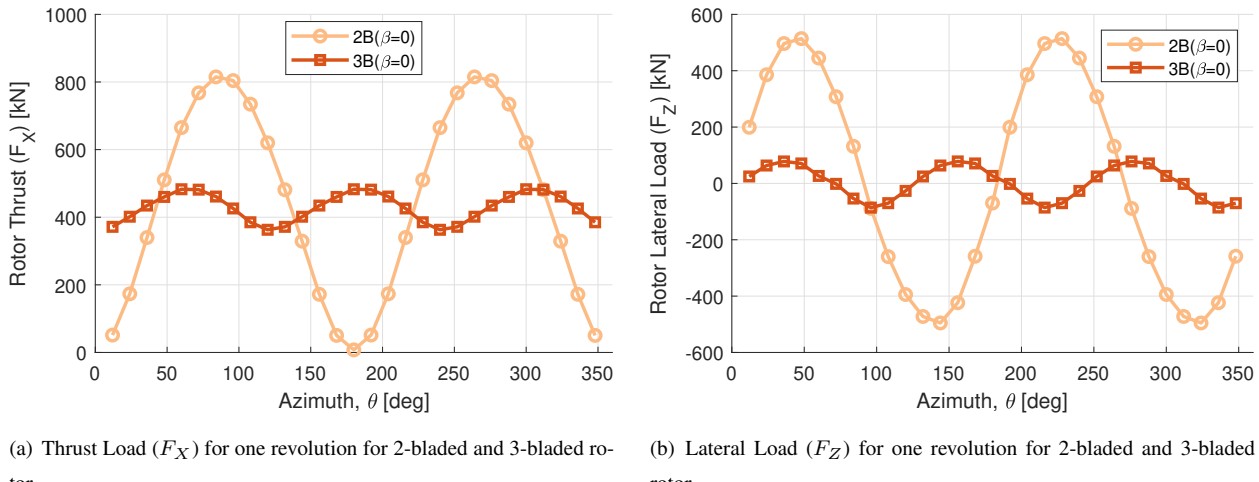

(a) Thrust Load ($F_X$) for one revolution for 2-bladed and 3-bladed rotor.

(b) Lateral Load ($F_Z$) for one revolution for 2-bladed and 3-bladed rotor.

**Figure 16.** Thrust Load ($F_X$) and Lateral Load ($F_Z$) for one revolution for 2-bladed and 3-bladed turbine.

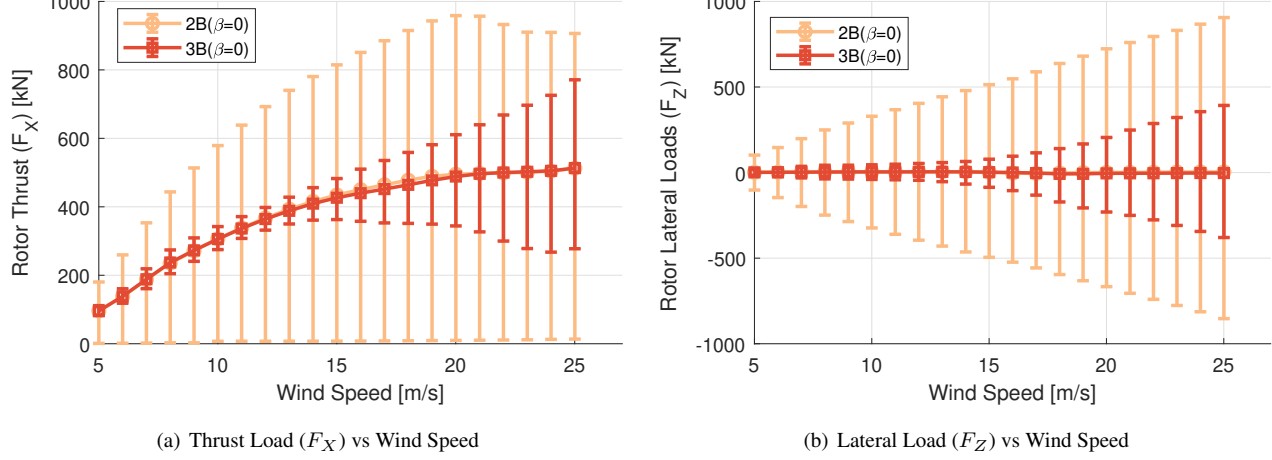

(a) Thrust Load ($F_X$) vs Wind Speed

(b) Lateral Load ($F_Z$) vs Wind Speed

**Figure 17.** Thrust load ($F_X$) and Lateral load ($F_Z$) during operating conditions.

### 3.2.3 Parked Loads

To demonstrate the effect of number of blades on parked loads it is important to show the orientation of loads to wind direction. The dependence of parked (thrust and lateral) loads on number of blades is demonstrated in Fig. 12 and Fig. 13. Wind direction

305   is plotted in the x-axis whereas the respective loads (thrust and lateral) are plotted on the y-axis and these trends for the six design iterations as mentioned previously have been shown. It is clear that the parked loads (thrust and lateral) are heavily dependent on wind direction for the 2-bladed rotor which is in direct contrast to 3-bladed case where the force is almost constant with wind direction. This is consistent with the findings from Ottermo et al. (2012). A reduction in these parked ripple





effects due to the addition of a third blade is similar to the results found in turbine loads change due to addition of a blade. The
maximum and minimum of parked thrust and lateral loads have been quantified in Table 3.

The change in cyclic load amplitude has been quantified according to previous sections. For $\beta = 0, 0.5$ & 1 the thrust ripple
effect decreases by 83.37 %, 82.75 % and 81.9% respectively. Similarly for lateral loads the ripple effect decreases by 77.66
%, 77.78 % and 77.9% respectively.

**Table 3.** Maximum and minimum of parked thrust ($F_X$) and lateral ($F_Z$) loads

| $\beta$ [$-$] | Thrust [$kN$] (Max. and Min.) 2B | Thrust [$kN$] (Max. and Min.) 3B | Lateral Load [$kN$] (Max. and Min.) 2B | Lateral Load [$kN$] (Max and Min.) 3B |
|---|---|---|---|---|
| 0 | 833.5 -16.2 | 577.4 436.1 | ± 639.8 | ± 143.0 |
| 0.5 | 993.6 -21.0 | 701.1 526 | ± 797.7 | ± 177.2 |
| 1 | 1094.0 24.8 | 788 585.5 | ± 912.2 | ± 201.5 |

### 3.3 Case 3: Impact of Aspect Ratio

As noted, there are multiple parameters affecting the performance of a VAWT. Thus far, we have examined solidity including
solidity changes through variable chord tapering and the impact of the number of blades. We now turn our attention to the
impact of aspect ratio (AR) and its effect on aerodynamic performance and loading. The effect of the changing rotor aspect
ratio (AR) on the aerodynamic forces generated by a VAWT and the performance coefficients will be discussed in this section.
As changes in aspect ratio (AR) affects the aerodynamics of a VAWT especially for Darrieus configurations, it is important to
study the change in performance and loads. For this study, aspect ratio is defined as the ratio between the rotor height (H) and
rotor diameter (D).

The effect of aspect ratio on VAWT performance have been investigated previously through both experiments and numerical
simulation and mostly for H-VAWT type rotors. The effect of changing the aspect ratio on H-type VAWTs has been studied by
Brusca et al. (2014) where a multiple stream tube model was used. The authors investigated the Reynolds number relation with
changing aspect ratio and concluded that a decrease in aspect ratio increases efficiency. However, this increase in performance
can be ascribed to mainly two factors. First, as the designed turbine under consideration is a small-scale turbine (1kW) all the
simulations are in low Reynolds number regime where the power coefficient is heavily dependent on the operating Reynolds
number (Miller et al. (2018); Armstrong et al. (2012)). Secondly, since the aspect ratio was varied by changing the rotor
radius, this causes a change in the solidity or chord to radius ratio ($c/R$) of the turbine and thus resulting in increased turbine
performance coefficient. As a result it would be erroneous to link increase in performance to lower aspect ratio as other
parameters (radius, solidity) are not being kept constant through the study.





The effect of aspect ratio on H-type VAWTs also has been studied using a 3D panel method by Maeda et al. (2017). In this study, analysis was done for a range of changing ratio (0.4 to 1.2) of the diameter (D) and blade span length (H) at a fixed solidity. The outcome of this study is that with increasing ratio of diameter and blade span length ($H/D$) there is an increase

in turbine power coefficient ($C_P$) and this was attributed to lower tip losses with growing $H/D$ ratio and decreased circulation amount ratio for a wide range of blade span for small values of $H/D$.

Hunt et al. (2020) experimentally examined the performance of H-type VAWTs with blade end struts by varying the aspect ratio in the range of (0.95 to 1.63) and keeping all the other parameters constant. By keeping other non-dimensional parameters constant they were able to isolate the effect of aspect ratio on turbine performance. The conclusion of the study is that if blade

support structure (struts) losses are accounted for the rotor performance is invariant to change in aspect ratio.

Despite these studies of AR for H-type VAWT rotors, no similar studies of aspect ratio impacts on aerodynamic loads and power performance has been performed for Darrieus-type VAWT rotors. In the following sections, we employ the CACTUS code in a 3D analysis of these aerodynamic effects for change in aspect ratio similar to the prior studies for variable blade chord tapering and number of blades shown in prior sections.

### 3.3.1   Turbine Performance

As seen from the literature that different non-dimensional parameters can give us erroneous perceptions regarding the effect of aspect ratio on rotor performance; therefore, in this study the aspect ratio has been varied in such a manner that the rotor diameter (D) and solidity remain constant so that only the rotor height (H) is changed. The aspect ratio (AR) study has been conducted for both 2-bladed (2B) and 3-bladed (3B) turbines, for $\beta = 0$, 0.5 and 1 and for AR = 0.5, 0.75, 1, 1.5 as listed in

Table 1.

Fig. 18 shows the change in power coefficient ($C_P$) and TSR for different AR values and $\beta = 0$. On the x-axis we have TSR and on y-axis we have power coefficient ($C_P$) and they have been plotted for both 2B and 3B turbines. $\beta = 0.5$ and 1 are intentionally left out, as similar trends are observed as for $\beta = 0$.

As seen from the figure the peak of power coefficient increases with the increase of aspect ratio (H/D). But the optimum

TSR also increases with the increase of H/D. The maximum values of power coefficient ($C_P$) for both 2B ($\beta = 0$) and 3B ($\beta = 0$) turbines has been noted in Table 4.

**Table 4.** Maximum power coefficient ($C_P$) for different aspect ratio (AR) rotor.

| $\beta$ [$-$] | Aspect Ratio ($H/D$) [$-$] | Power Coefficient ($C_P$) [$-$] 2B | Power Coefficient ($C_P$) [$-$] 3B |
|---|---|---|---|
| 0 | 0.5 | 0.3226 | 0.3231 |
|  | 0.75 | 0.3698 | 0.3711 |
|  | 1 | 0.3975 | 0.3986 |
|  | 1.5 | 0.4250 | 0.4264 |





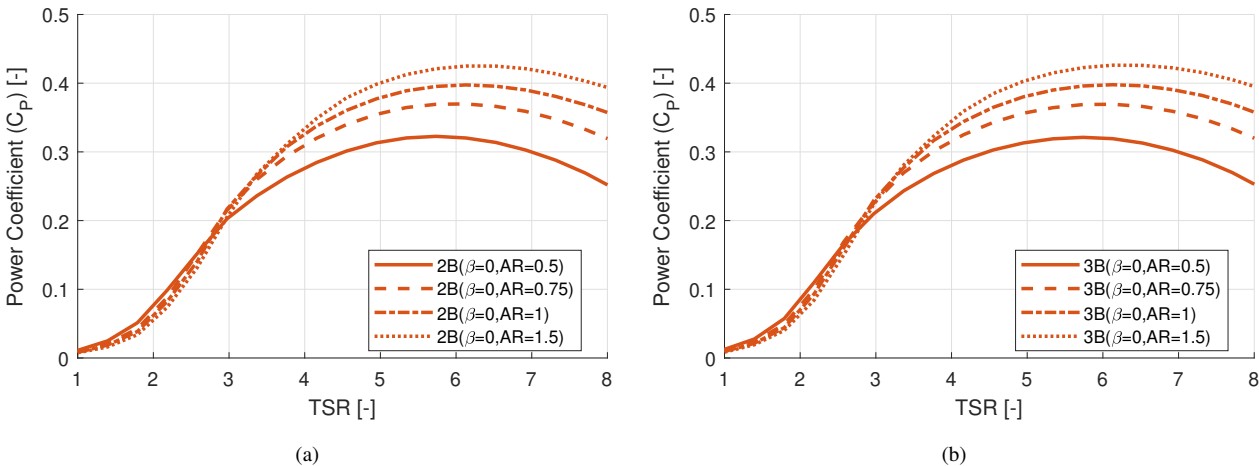

**Figure 18.** Power Coefficient ($C_P$) vs TSR for different aspect ratio (AR) rotor.

This increase in power coefficient with aspect ratio (AR) can be attributed to blade curvature effects. As low H/D ratio turbine has more blade curvature compared to a high H/D ratio turbine, this limits the region of blade span producing high torque for low H/D ratio turbine. To demonstrate this effect, the power coefficient ($C_P$) of the individual blades for the 2-bladed

(2B, $\beta$ =0) for all aspect ratio (AR) values under consideration has been plotted against the rotor azimuth angle (Fig. 19). We can see that with increasing aspect ratio each blade has greater contribution to rotor power coefficient and this due to the fact that higher aspect ratio rotor has greater blade span producing higher torque values.

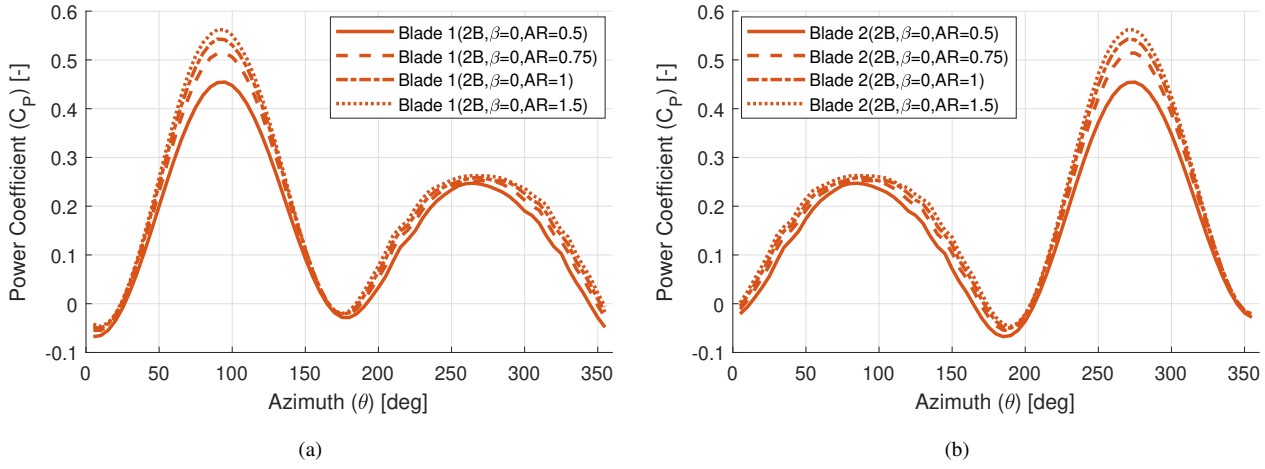

**Figure 19.** Individual blade power coefficient ($C_P$) for different aspect ratio (AR) rotor.





### 3.3.2 Turbine Loads

The effect of increasing the aspect ratio (AR) on the thrust and lateral loads are studied in this section. Fig. 20 shows the effect

of changing the aspect ratio on the thrust coefficient ($C_x$) and Fig. 21 shows the effect of changing the aspect ratio on the lateral load coefficient ($C_z$). As expected, it can be seen from the Fig. 20 that with increasing aspect ratio (AR) and TSR the turbine thrust coefficient ($C_x$) goes up too. The trends remain same for 3-bladed (3B) turbines too as solidity is kept constant across 2-bladed (2B) and 3-bladed (3B) designs. For lateral loads the mean value is not affected by a change in aspect ratio (AR) but the absolute value of the maximum and minimum lateral loads increase with AR and TSR (Fig. 21). Again we see similar

behavior of lateral loads for 3-bladed (3B) turbines, where adding another blade significantly reduces the range of the loads.

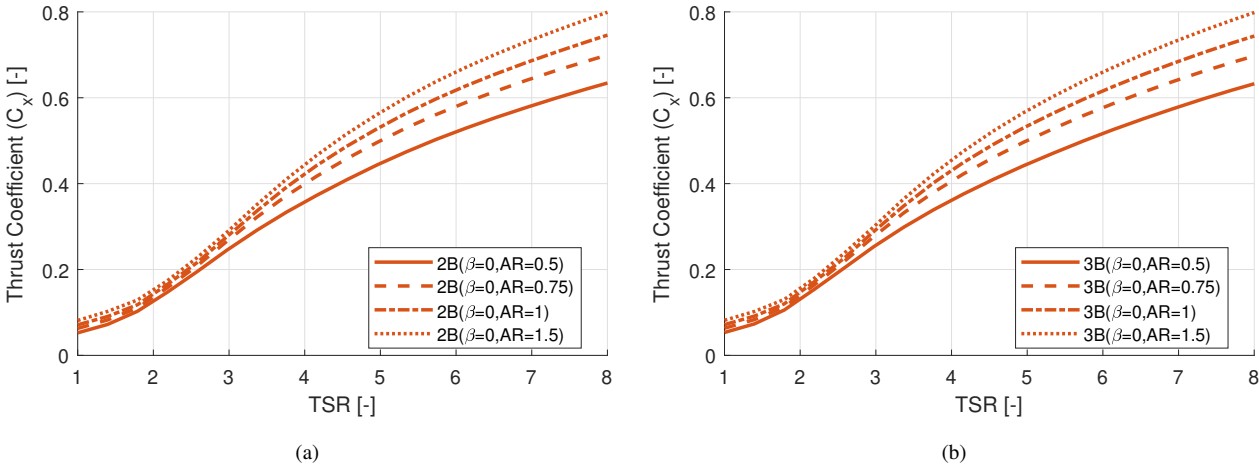

**Figure 20.** Thrust coefficient ($C_x$) vs TSR for different aspect ratio (AR) rotors.

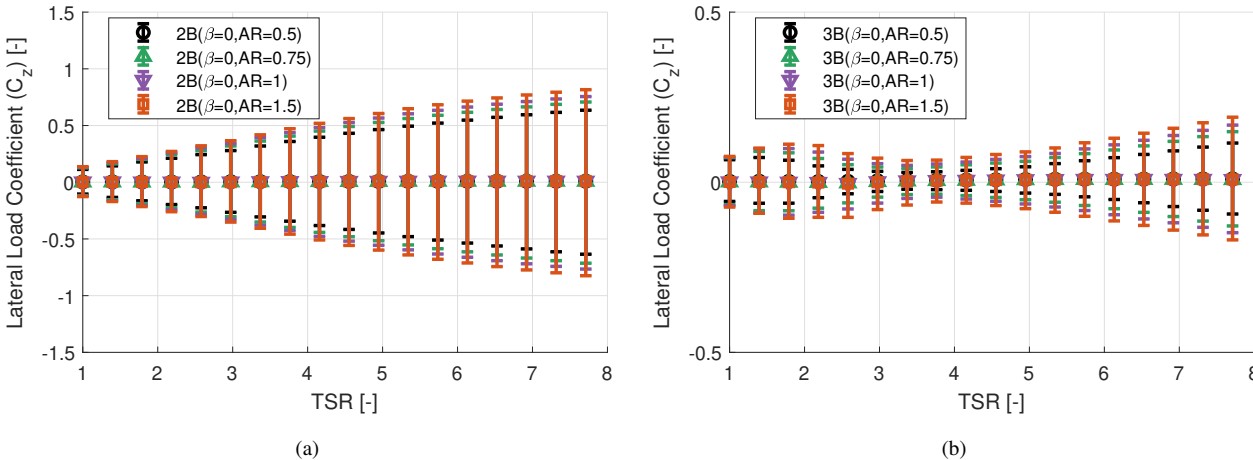

**Figure 21.** Lateral load coefficient ($C_z$) vs TSR for different aspect ratio (AR) rotors.



### 3.3.3 Parked Loads

Because parked loads are a major load case in designing VAWTs (Griffith et al. (2018)), it is important to understand the relation between change in aspect ratio (AR) and parked loads. Fig. 22 and 23 shows the variation of parked (thrust and lateral) loads for different aspect ratios for 2-bladed (2B) and 3-bladed (3B) turbines. Plots have been shown only for $\beta = 0$ case as the trends remain similar for the other $\beta$ values. It is observed that with an increase of aspect ratio the both the thrust and lateral loads increases, which is consistent to the findings of effect of aspect ratio on turbine loads (previous section). Adding another blade significantly reduces the ripple effect and makes the loads independent of wind direction.

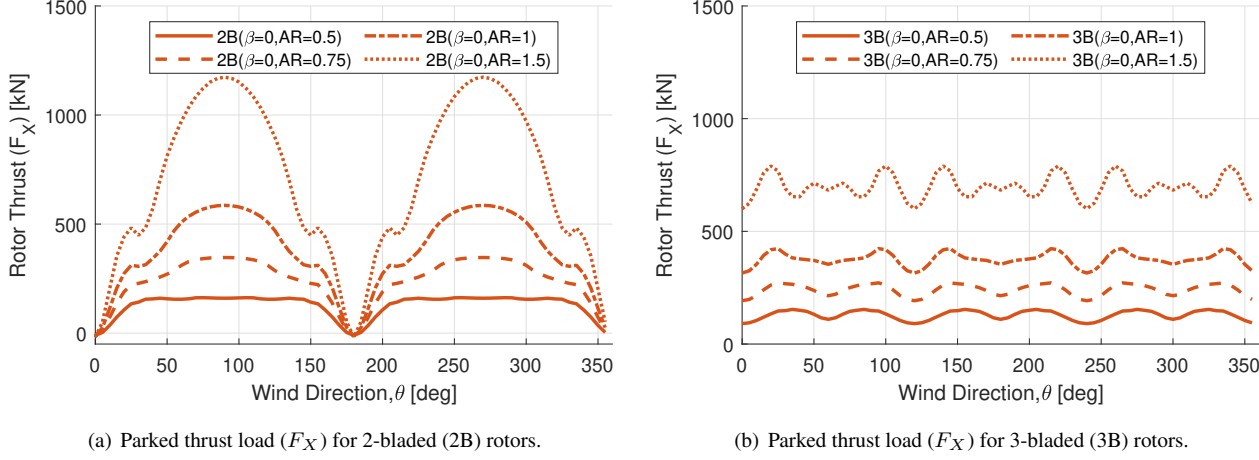

(a) Parked thrust load ($F_X$) for 2-bladed (2B) rotors.   (b) Parked thrust load ($F_X$) for 3-bladed (3B) rotors.

**Figure 22.** Parked rotor thrust load ($F_X$) analysis for different aspect ratio (AR) rotors.

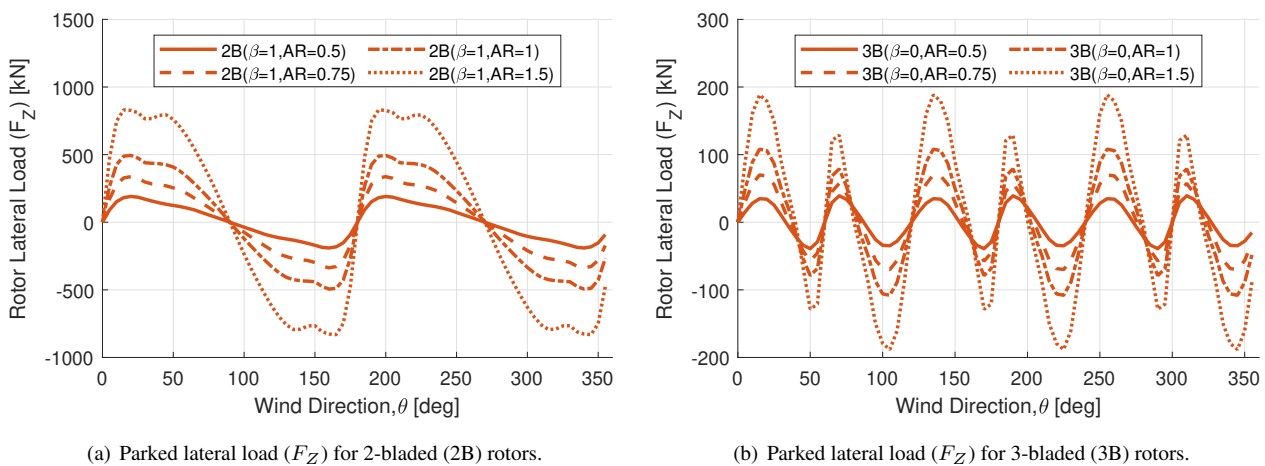

(a) Parked lateral load ($F_Z$) for 2-bladed (2B) rotors.   (b) Parked lateral load ($F_Z$) for 3-bladed (3B) rotors.

**Figure 23.** Parked rotor lateral load ($F_Z$) analysis for different aspect ratio (AR) rotors.



# 4 Conclusions

In order to generate modern and efficient VAWT rotor designs one needs to understand the design space and the variables
affecting the aerodynamic design space of a VAWT. However to capture the effect of one design variable on the turbine
performance it is important to isolate the effect of that particular design variable so we can reach proper conclusions. In this
work a comprehensive and systematic study of the effect of blade tapering ($\beta$), number of blades (N) and aspect ratio (AR)
on Darrieus-type VAWT aerodynamic performance has been conducted using a vortex model named CACTUS to evaluate the
power coefficient, and the instantaneous thrust and lateral forces. As mentioned previously parked loads are an important load
case for VAWTs due to lack of pitching mechanism and the omnidirectional nature of VAWTs. So a comprehensive analysis of
parked loading on VAWTs have been studied too. In order to get representative power curves and operating loads of a turbine
an operational strategy development procedure for stall controlled turbines was discussed too.

In this study, a mid-fidelity 3D vortex model named CACTUS (which has been validated with experimental data) has been
used to carry out all the analysis. Both 2-and 3-bladed turbines have been studied for different blade chord tapering schemes
($0 \leq \beta \leq 1$) and different aspect ratios ($0.5 \leq AR \leq 1.5$). It is important to note that for a particular $\beta$ value turbine 2B and
3B turbines have the exact same solidity. This study also covers a range of tip speed ratios ($1 \leq TSR \leq 8$) and full operational
wind speeds (5-25) m/s.

Some of the highlights of the present study are as follows:

1. Blade Chord Tapering Impacts:

    (a) As tapering has been applied to the blades by adding solidity, the power coefficient ($C_P$) increases with increasing
        tapering but the optimum TSR decreases.

    (b) With an increase in tapering the maximum operating RPM to produce a rated power of 5MW also decreases and
        lower RPM leads to higher torque, which will increase the cost of the drivetrain.

    (c) As for turbine loads there is an increase in thrust coefficient with increasing $\beta$ value, and for lateral load coefficient
        the mean remains almost unaffected by the $\beta$ value. However due to the symmetric nature of lateral loads the peak
        ( max. and min. ) during a single revolution increases due to increase in $\beta$ and TSR values.

    (d) We also see similar trends for parked loading too. With increase in $\beta$ the peak of rotor thrust and lateral parked
        loads increase as well.

2. Number of Blades Impacts:

    (a) The effect of number of blades on power coefficient has been studied both at low and high wind speeds to observe
        the effect of Reynolds Number on trends since changing number of blades directly affects chord to radius ($c/R$) of
        the rotor. At low wind speeds the 2B rotors slightly outperform the 3B rotors especially at low TSR region and at
        sufficiently high wind speeds the power coefficient becomes insensitive to changes in number of blades





(b) As the solidity is kept constant for both 2-bladed and 3-bladed rotors for a particular $\beta$ value, the thrust coefficient
is also insensitive to number of blades.

(c) Adding one blade to 2B turbines significantly affects the range of the load profiles or the ripple effect. For example
when one blade is added to 2B ($\beta = 0$) turbine in such a way that solidity is kept constant the range of the thrust
and lateral loads reduces by 85.16 % and 83.76 % respectively at a single wind speed. These reductions can also
be observed for the whole range of operating wind speed (5-25)m/s.

(d) Similar trends are also seen for parked loads variation. There is a massive reduction in range of both thrust and
lateral parked loads for the 3-bladed rotor in comparison to 2-bladed rotor. It is important to note that the magnitude
of parked loading on a 3-bladed turbine is independent to the incoming wind direction which is in direct contrast to
a 2-bladed turbine. So parked loading on a 2-bladed turbine can be significantly reduced through optimal parking
operational strategy.

3. Aspect Ratio Impacts:

(a) The effect of changing aspect ratio (AR) has been quantified for (AR = 0.5,0.75,1 and 1.5) and is defined by $H/D$.
The aspect ratio was changed in such a way the the rotor radius or diameter remains constant while the rotor height
keeps changing.

(b) With increase in aspect ratio the power coefficient, thrust coefficient and peaks of lateral load coefficient increases.

(c) A similar trend is observed for parked loads too where the peak of thrust and lateral loads increases with increase
in aspect ratio.

The findings in this work provides a better understanding of the effect of blade chord tapering, rotor aspect ratio and number
of blades on the power, thrust, lateral and parked loads and shed light on some important load conditions which are usually
overlooked, namely VAWT parked loads and the cyclic lateral (side-to-side) loading components. In future studies to better
quantify the effects of these design variables the analysis should be coupled with a levelized cost of energy (LCOE) analysis.
This can be really insightful as a small change in a single parameter will change the cost of energy as other variables are also
affected. For example to see if the cost of adding an extra blade is lower than the savings it would lead to or if opting for a
turbine with higher $\beta$ will lead to cost savings or added expenditure due to larger drivetrain. In addition, all the analysis has
been carried out for a single airfoil. A more comprehensive analysis can be performed with much modern airfoils tailored
specifically for VAWTs. Also as dynamic stall is a major concern for VAWTs more detailed analysis can be carried out with
dynamic stall models tuned for a particular rotor case that can give one a more realistic picture of VAWT operating loads and
performance.



*Author contributions.* This work was performed during the M.Sc. of MSS under the supervision of DTG as part of a Advanced Research Projects Agency–Energy (ARPA-E) funded project named 'A Low-cost Floating Offshore Vertical Axis Wind System'. MSS and DTG
contributed to the analysis and interpretation of the data and the manuscript was prepared by MSS with the help of DTG.

*Competing interests.* The authors declare no conflict of interest.

*Acknowledgements.* The research presented herein was funded by the US Department of Energy Advanced Research Projects Agency-Energy (ARPA-E) under the ATLANTIS program with project title "A Low-cost Floating Offshore Vertical Axis Wind System" with Award No. DE-AR0001179. Any opinions, findings, and conclusions or recommendations expressed in this material are those of the authors and do
not necessarily reflect the views of ARPA-E. The authors are grateful for the support of the ARPA-E program and staff, and the project team.



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
