# Peer review of "Parked and Operating Loads Analysis in the Aerodynamic Design of Multi-megawatt-scale Floating Vertical Axis Wind Turbines"

_Wind Energy Science, 2021_

## Referee Comment (RC1)

**Reviewer's comments to manuscript no. WES-2021-60**

Manuscript no.:    WES-2021-60

Full title:    Parked and Operating Loads Analysis in the Aerodynamic Design of Multi-megawatt-scale Floating Vertical Axis Wind Turbines

Authors:    Mohammad Sadman Sakib and D. Todd Griffith

The manuscript deals with the analysis of multi-megawatt-scale floating Darrieus-type vertical axis wind turbines. Both parked and operating conditions are evaluated by means of performance (i.e., power extraction) and rotor-averaged aerodynamic loads. A 3D vortex model was used to perform the simulations. Different design configurations were analyzed, with variations in number of blades, aspect ratios (ratios of rotor height and diameter), and blade tapering towards the blade ends. During the design variations with respect to the number of blades and aspect ratios, the authors took care not to modify the rotor solidity and Reynolds numbers, which yields clear conclusions that are affected by one parameter only and not a mixture of parameters as was done by other authors before.

**General Comments**

The manuscript is generally written in a clear and concise manner and well organized. The ideas and messages are clearly formulated and the overall story-telling is very good. Here and there, a nice systems engineering view shines through that is very appealing. The topic of the manuscript is generally relevant for the wind energy research community. The methods for the description of the turbine aerodynamics are appropriate for a design space analysis. The results are interesting and allow for clear conclusions provided at the end of the manuscript. There are just a few specific comments and technical corrections that should be addressed before final acceptance of the manuscript, which are given below.

**Specific Comments**

In the introduction, there are a number of self-citations which yields the impression that there are no other research groups working in the field of vertical axis wind turbines. The authors may find additional references to broaden the view on available literature. Also, the authors state that "various floating VAWT concepts have been proposed" (p. 2, l. 36), but give only two examples. The authors may give additional references to existing concepts.

The authors evaluate loads in a rotor-averaged manner, i.e., rotor thrust and lateral forces are analyzed. It would be interesting to see the difference in local loads on the blades depending on the variation of design parameters, e.g., by means of blade root bending moments. A respective extension would be appreciated, as it would allow for implications for structural blade design, which in turn would also underline the systems engineering view visible in some parts of the manuscript.

On page 12, at the end of the first paragraph, the authors list the simplifications and limitations of the model applied for the calculation of standstill situations, which is good. However, these seem quite numerous, and it is not entirely clear enough why they have been included. The authors should thus elaborate on the necessity to consider the simplifications.

The argumentation on page 12, 3rd paragraph, is weak. What differences in the overall theory behind the calculation of parked loads are the reason for the differences in the overall shape of the profiles? Please be more specific.

**Technical Corrections**

Please avoid unnecessary abbreviations in the abstract (N, AR, H, D). The authors should further consider restructuring section 2. Subsections and subsubsections may not be necessary, if there is only one subsection and subsubsection.

---

## Referee Comment (RC2)

Review of Manuscript wes-2021-60 MS

Title: Parked and Operating Loads Analysis in the Aerodynamic Design of Multi-megawatt-scale Floating Vertical Axis Wind Turbines
Author(s): Mohammad Sadman Sakib and D. Todd Griffith MS
Type: Research article

**Summary**

As in the abstract stated, this paper describes load calculations with a vortex code for Vertical Axis wind turbines. The manuscript is clearly written and the results are as expected (a three-bladed version shows less variation than a 2-bladed one) To my opinion, the manuscript is a little bit too extended. Authors should consider if it can be shortened.

**General**

The approach using a vortex type code to cover a broader parameter range seems meaningful as well as to focus on stand-still conditions. To get more insight on the overall accuracy of the "medium fidelity" approach authors should refer to recent CFD work, for example
Bangga et al. Energy 206 (2020) 118087

**Specific**

Figure 11: the information might be easier to compare if cT (thrust coefficient) instead of absolute force would have been presented.

As the investigations presented seem to be part of a larger project "A Low-cost Floating …" it would be interesting to reads about how far this goal was achieves

**Typos**

Line 106:  2nd dot superfluous; exponent (-5) instead of 5

---

## Author Comment (AC1)

**Title: Parked and Operating Loads Analysis in the Aerodynamic Design of Multi-megawatt-scale Floating Vertical Axis Wind Turbines [https://doi.org/10.5194/wes-2021-60]**
**Author(s): Mohammad Sadman Sakib and D. Todd Griffith**

Thank you for your feedback which is extremely helpful. The author responses (**in blue**) to the reviewer comments (in black) are noted below.

The manuscript deals with the analysis of multi-megawatt-scale floating Darrieus-type vertical axis wind turbines. Both parked and operating conditions are evaluated by means of performance (i.e., power extraction) and rotor-averaged aerodynamic loads. A 3D vortex model was used to perform the simulations. Different design configurations were analyzed, with variations in number of blades, aspect ratios (ratios of rotor height and diameter), and blade tapering towards the blade ends. During the design variations with respect to the number of blades and aspect ratios, the authors took care not to modify the rotor solidity and Reynolds numbers, which yields clear conclusions that are affected by one parameter only and not a mixture of parameters as was done by other authors before.

Response: Thank you for your comments which are greatly appreciated.

**General Comments**

The manuscript is generally written in a clear and concise manner and well organized. The ideas and messages are clearly formulated and the overall story-telling is very good. Here and there, a nice systems engineering view shines through that is very appealing. The topic of the manuscript is generally relevant for the wind energy research community. The methods for the description of the turbine aerodynamics are appropriate for a design space analysis. The results are interesting and allow for clear conclusions provided at the end of the manuscript. There are just a few specific comments and technical corrections that should be addressed before final acceptance of the manuscript, which are given below.

Response:  Thank you again for your valuable feedback. We have tried our best to respond your specific comments as below.

**Specific Comments**

In the introduction, there are a number of self-citations which yields the impression that there are no other research groups working in the field of vertical axis wind turbines. The authors may find additional references to broaden the view on available literature. Also, the authors state that "various floating VAWT concepts have been proposed" (p. 2, l. 36), but give only two examples. The authors may give additional references to existing concepts.

Response: The authors acknowledge the fact that there are several other research teams working on interesting problems related to VAWTs. We addressed this issue in the revised manuscript by including additional references to ongoing VAWT work and different floating VAWT concepts proposed, as noted on Page 1 Line 20-24 and Page 2 Line 38-39.

The authors evaluate loads in a rotor-averaged manner, i.e., rotor thrust and lateral forces are analyzed. It would be interesting to see the difference in local loads on the blades depending on the variation of design parameters, e.g., by means of blade root bending moments. A respective extension would be appreciated, as it would allow for implications for structural blade design, which in turn would also underline the systems engineering view visible in some parts of the manuscript.

The authors agree that more discussion of aerodynamic design and loads and its implications on other aspects of the systems engineering like structural design would greatly benefit the manuscript. Such additions are made as mentioned: Page 14 Line (256-261), Page 18 Line (332-340), and Page 23 Line (405-408).

On page 12, at the end of the first paragraph, the authors list the simplifications and limitations of the model applied for the calculation of standstill situations, which is good. However, these seem quite numerous, and it is not entirely clear enough why they have been included. The authors should thus elaborate on the necessity to consider the simplifications.

Response: The parked loads or standstill loads can be a major concern in VAWTs. The model implemented in this work has some limitations like no wake effect consideration and one blade element does not affect another but these are reasonable assumptions for standstill loading analysis. These assumptions may result in some discrepancies in experimental and simulated loads and as a result, the simplifications and limitations have been explicitly highlighted on Page 12, Line 225-229. It is important to note that as a lot of design iterations have to be completed in the initial stage of turbine design this vortex-based model proves very useful and also serves as a good numerical tool for rotor parameter optimization too.

The argumentation on page 12, 3rd paragraph, is weak. What differences in the overall theory behind the calculation of parked loads are the reason for the differences in the overall shape of the profiles? Please be more specific.

Response: Ottermo et al. (2012) manuscript details the simplified relation used between standstill thrust loading and wind velocity which is shown below.

$$F(\theta) = \frac{S\rho v^2}{2N} \sum_i C_D\left(\theta + \frac{2\pi i}{N} + \alpha\right)$$

Here S is Blade area, v is wind speed, N is number of blades, i =1, 2 …N is the blade label, $C_D$ is drag coefficient and $\theta$ is azimuth angle. It is clearly seen that in the equation thrust load is only a function of drag coefficient and the azimuth angle and does not take into account the lift coefficient ($C_L$). In actuality thrust loads are function of both normal and tangential load components which are in turn functions of lift and drag coefficients. These lift and drag coefficients are obtained based on the nominal angle of attack of blade elements which is again a function of normal and tangential components of velocity. In the current study the thrust and lateral loads are functions of both $C_L$ and $C_D$ which results in difference of shape of profiles.

**Technical Corrections**

Please avoid unnecessary abbreviations in the abstract (N, AR, H, D). The authors should further consider restructuring section 2. Subsections and subsubsections may not be necessary, if there is only one subsection and subsubsection.

Response: We have removed the abbreviations from the abstract. Section 2 is restructured to contain only two subsections rather than one subsection and one subsubsection.

---

## Author Comment (AC2)

**Title: Parked and Operating Loads Analysis in the Aerodynamic Design of Multi-megawatt-scale Floating Vertical Axis Wind Turbines [https://doi.org/10.5194/wes-2021-60]**
**Author(s): Mohammad Sadman Sakib and D. Todd Griffith**

Thank you for your valuable feedback which is extremely helpful. The author responses (**in blue**) to the reviewer comments (in black) are noted below.

**Summary**
As in the abstract stated, this paper describes load calculations with a vortex code for Vertical Axis wind turbines. The manuscript is clearly written and the results are as expected (a three-bladed version shows less variation than a 2-bladed one) To my opinion, the manuscript is a little bit too extended. Authors should consider if it can be shortened.

Response: Considering the various design variables we are studying (number of blades, tapering, aspect ratio) and their effect on turbine performance, including both operating and parked loads, the resulting manuscript is plot heavy. We have tried our best to ensure the manuscript reads well without extending the discussion too much.

**General**
The approach using a vortex type code to cover a broader parameter range seems meaningful as well as to focus on stand-still conditions. To get more insight on the overall accuracy of the "medium fidelity" approach authors should refer to recent CFD work, for example
Bangga et al. Energy 206 (2020) 118087

Response: The authors agree that the manuscript would benefit from referencing some high fidelity (CFD) work and vortex model compares against them. Thus, we have added a small extension on page 3, line 66-73.

**Specific**
Figure 11: the information might be easier to compare if cT (thrust coefficient) instead of absolute force would have been presented.

Response: As we don't have access to the original model that was used in Ottermo et al. (2012), the thrust coefficients of the turbine used could not be obtained. So, an online tool called webplot digitizer was used to extract the original data points from the plot in the manuscript and subsequently compared with the current model.

As the investigations presented seem to be part of a larger project "A Low-cost Floating …" it would be interesting to reads about how far this goal was achieves

Response: This is still an ongoing project and currently we do not have any published data on levelized cost of energy of the floating platform VAWT system. However, the impact of aerodynamic loads and on the floating platform design has been published in the journal 'Renewable Energy' which can be found at (https://doi.org/10.1016/j.renene.2021.09.076) [1]. Also effect of blade tapering on aeroelastic stability (flutter) for floating VAWTs has been recently accepted by the journal 'Renewable Energy' and can be found at [https://doi.org/10.1016/j.renene.2021.12.041] [2]. More details on ongoing and future work related to the project can be found at the project website: https://labs.utdallas.edu/griffith/.

**Typos**
Line 106: 2nd dot superfluous; exponent (-5) instead of 5

Response: Corrected and units added.

REFERENCES:

1.  Gao, Ju, et al. "A semi-coupled aero-servo-hydro numerical model for floating vertical axis wind turbines operating on TLPs." *Renewable Energy* 181 (2022): 692-713.
2.  Ahsan, Faraz et al. "Modal dynamics and flutter analysis of floating offshore vertical axis wind turbines " *Renewable Energy [Accepted]*, 2021, [https://doi.org/10.1016/j.renene.2021.12.041]